# Novel Biomarkers and Advanced Cardiac Imaging in Aortic Stenosis: Old and New

**DOI:** 10.3390/biom13111661

**Published:** 2023-11-17

**Authors:** Anca Drăgan, Anca Doina Mateescu

**Affiliations:** 1Department of Cardiovascular Anaesthesiology and Intensive Care, Emergency Institute for Cardiovascular Diseases “Prof Dr C C Iliescu”, 258 Fundeni Road, 022328 Bucharest, Romania; 2Department of Cardiology, Emergency Institute for Cardiovascular Diseases “Prof Dr C C Iliescu”, 258 Fundeni Road, 022328 Bucharest, Romania; ancad.mateescu@gmail.com

**Keywords:** aortic stenosis, myocardial fibrosis, cardiac biomarkers, remodeling, risk assessment

## Abstract

Currently, the symptomatic status and left ventricular ejection fraction (LVEF) play a crucial role in aortic stenosis (AS) assessment. However, the symptoms are often subjective, and LVEF is not a sensitive marker of left ventricle (LV) decompensation. Over the past years, the cardiac structure and function research on AS has increased due to advanced imaging modalities and potential therapies. New imaging parameters emerged as predictors of disease progression in AS. LV global longitudinal strain has proved useful for risk stratification in asymptomatic severe AS patients with preserved LVEF. The assessment of myocardial fibrosis by cardiac magnetic resonance is the most studied application and offers prognostic information on AS. Moreover, the usage of biomarkers in AS as objective measures of LV decompensation has recently gained more interest. The present review focuses on the transition from compensatory LV hypertrophy (H) to LV dysfunction and the biomarkers associated with myocardial wall stress, fibrosis, and myocyte death. Moreover, we discuss the potential impact of non-invasive imaging parameters for optimizing the timing of aortic valve replacement and provide insight into novel biomarkers for possible prognostic use in AS. However, data from randomized clinical trials are necessary to define their utility in daily practice.

## 1. Introduction. Current State of Knowledge

The prevalence of valvular diseases is increasing with the prediction of by 2050 [1], especially in developed countries. The aging population, the increasing availability of imaging techniques, and accessibility to diagnosis and treatment may be some reasons for this trend [2]. The prevalence of AS rises sharply with increasing age and approaches 25% of all adults older than 65 years [3], with 2–5% of patients having severe AS [3]. In the UK, one study reported a prevalence of 1.48% for severe AS with 31.7% asymptomatic cases for people aged ≥55 years [4]. Stewart et al. reported that 21% of men and 18.7% of women aged 65 years or older in Australia had evidence of mild-to-severe AS [5]. A percentage of 2.3% men and 1.9% women presented severe AS. They also demonstrated that not only severe AS but also mild and moderate AS might significantly reduce the longevity and quality of life of people aged 65 years or older [3,5].

This review focuses on the transition from compensatory LVH to LV dysfunction and heart failure in AS and the release of biomarkers associated with myocardial wall stress, fibrosis, and myocyte death. Moreover, we discuss the potential impact of non-invasive imaging parameters for optimizing the timing of aortic valve replacement (AVR) and provide insight into novel biomarkers for possible diagnostic and prognostic use in AS.

Currently, the symptomatic status and left ventricular ejection fraction (EF) play a crucial role in the management of AS patients [6,7]. However, the symptoms are often subjective, and LVEF is not a sensitive marker of LV decompensation. As a consequence, the timing of aortic valve operations is often suboptimal. Research related to the cardiac structure and function in patients with AS has increased over the past years due to advanced imaging modalities and potential therapies, in particular the emergence of transcatheter aortic valve implantation (TAVI).

The progressive decrease in the area of the aortic valve determines a chronic pressure overload. The early adaptive manifestation of the ventricular myocardium is hypertrophy with LV diastolic dysfunction. Later, LV systolic dysfunction occurs, with myocardial contractile function and deformation becoming impaired. The left atrium (LA) also presents some morphological changes. The LA responds by increasing its volume to the chronic pressure overload, leading to a higher pulmonary venous and arterial pressure. The ventricular and atrial changes will result in heart failure, atrial fibrillation (AF), and symptom development. Thus, AS represents not only a valvular disease but a whole heart disease, often in patients with comorbidities. However, AS patients can be asymptomatic until the late stage of the disease.

The presence of symptoms in severe AS patients was reported to be associated with smaller aortic valve area, higher degree of LVH, increased levels of plasmatic brain natriuretic peptide, but also with impaired LV diastolic function parameters, including increased LA dimensions [8,9,10,11,12].

Since there is no other definitive treatment apart from AVR in AS, the time of intervention is crucial to optimizing the outcome for patients and their future quality of life. While for symptomatic severe AS patients, the AVR decision is clearly stipulated, the timing for intervention in asymptomatic severe AS patients is still subject to debate. Studies already reported that symptom development is associated with higher mortality, even after AVR. Asymptomatic patients with moderate-to-severe AS demonstrated unequivocal progression in the adverse cardiac remodeling within 12 months, with a significant increase in focal myocardial fibrosis (MF) [13]. A multi-center cardiac magnetic resonance (CMR) study reported that in AS patients, cellular hypertrophy and diffuse fibrosis progressed rapidly, but this change was reversible after AVR [14].

Once established, mid-wall late gadolinium enhancement (LGE) also accumulated rapidly, increasing by 75% each year on average, especially in patients with a high baseline fibrosis burden [14]. This change was irreversible following valve replacement [14]. Therefore, it is clinically relevant to better define the optimal timing of valve intervention in AS. The answer may not be found in one single assessment method but in a multi-parametric approach, which integrates imaging and serum biomarkers related to both ventricular and atrial myocardium function and structure, focusing on the potential reversal of structural changes, such as MF.

## 2. Pathophysiology of Cardiac Dysfunction in Aortic Stenosis

### 2.1. Left Ventricular Response to Aortic Stenosis

Left ventricular hypertrophy (LVH) develops early, in response to pressure overload from AS. The LV remodeling pattern may vary between patients depending on sex, age, and co-existent coronary artery disease (CAD) or hypertension [15]. Although initially beneficial, the LV chronic hypertrophic response may be deleterious, with patients transitioning to symptoms, including heart failure, and adverse events. Stein et al. reported that LVH was independently associated with all-cause mortality in AS patients [16]. Myocyte degeneration, cell death, and fibrosis may be the structural changes responsible for this transition. The high myocardial oxygen demand is unbalanced in severe AS by the insufficient coronary capillary network, leading to impaired myocardial perfusion and cardiomyocyte cell death [17,18].

LVH and MF from chronically elevated LV systolic pressure result in diastolic LV dysfunction [19]. The gradual and often incomplete improvement in diastolic dysfunction (DD) follows LV remodeling after AVR [19]. Klein et al. draw attention to the timely detection of more advanced stages of DD in AS to identify the asymptomatic patients who would benefit from AVR, knowing that DD may be multi-factorial due to comorbidities, which impact diastolic LV function, such as hypertension and amyloidosis, particularly in elderly patients [20]. If the disease progresses, irreversible myocardial damage and interstitial fibrosis occur, leading to LV systolic dysfunction and a further decline in LVEF [21]. Stassen et al. recently demonstrated the importance of DD of LV in AS patients with preserved LVEF [22]. All-cause mortality was significantly dependent on DD of LV even in moderate AS [22].

The term fibrosis is used in the literature to describe the excessive deposition of extracellular matrix (ECM) proteins in parenchymal tissues and typically reflects inappropriate or unrestrained activation of a reparative program [23]. In AS, MF must be considered a dynamic process. To date, the mechanisms governing its development and progression in AS are incompletely understood. The contributors to the development of fibrosis in AS patients are an imbalance in matrix metalloproteinases and tissue inhibitors of matrix metalloproteinase (MMPs) activity, alongside increases in angiotensin-converting enzyme and transforming growth factor beta1 activity. Each of these mechanisms could be a potential target for aortic valve intervention [24,25].

In the early stages of the disease, the process involves the myocardium diffusely, is interstitial, reactive to pressure overload, and potentially reversible [26].

In the late stages of the disease, the fibrosis becomes substitutive and irreversible, with a focal distribution [26] due to the persistence of pressure overload. Progressive AS and LVH result in impaired myocardial blood flow, diminished coronary reserve, and compensatory vasodilation of the remaining vessels, with microvascular dysfunction and reduced capillary density ensuing. Thus, one driver of replacement fibrosis in AS was considered to be microvascular ischemia [15].

Frangogiannis identified the four mechanisms, which may induce the activation of the fibroblasts in heart failure associated with pressure overload [23]. Neurohumoral activation has an essential role in myofibroblast conversion [23,27]. The induction of matricellular proteins locally activates growth-factor-mediated signaling in fibroblasts, stimulating ECM protein synthesis [23,28,29]. The direct activation of mechanosensitive cascade and the release of inflammatory cytokines and growth factors by the stressed cardiomyocytes and immune cells contribute to fibroblast activation [23,30,31,32].

The gold standard for MF assessment is histological analysis obtained through endomyocardial biopsy, but non-invasive cardiac imaging may offer surrogate biomarkers [33].

### 2.2. Left Atrial Response to Aortic Stenosis

In AS, the increased pressure and wall stress are also present at the atrium level. The LA plays an important role in modulating LV filling and maintaining an optimal LV stroke volume, especially in patients with AS and LVH, through several different mechanisms. The LA acts as a reservoir during LV systole and isovolumic relaxation, filling with blood from the pulmonary veins; as a conduit during early LV diastole and diastasis, transferring blood into the LV via a small pressure gradient during early diastole and passively from the pulmonary veins during diastasis; as a booster pump during late LV diastole, contributing to LV stroke volume by 20–30% in normal subjects and significantly more when LV diastolic properties are impaired; and as a suction source, which refills itself in early systole [34]. LV diastolic dysfunction is an independent predictor of cardiovascular events in the general population, being associated with adverse outcomes [35]. The LA volume index is one of the four parameters currently recommended for the evaluation of LV diastolic function by echocardiography [36].

MF at the LA level is very important, leading to LA dysfunction with symptom occurrence and/or atrial fibrillation (AF). The remodeling process becomes irreversible if this LA fibrosis is extensive. Thus, early detection of LA dysfunction is mandatory for initiating specific therapeutic interventions [37].

The increased LA pressure and wall stretch lead to the renin–angiotensin–aldosterone system (RAAS) and leukocyte activation—the main pathways to atrial fibrosis and cardiomyocyte hypertrophy [38].

RAAS activation promotes the activation of hydrolysis phospholipase C (PLC), the activation of the nicotinamide adenine dinucleotide phosphate (NADPH) oxidase, the release of reactive oxygen species (ROS), and it regulates the expression of profibrotic factors (TGF, CTGF) [38]. PLC activation leads to intracellular Ca^2+^ overload and fibroblast proliferation [38]. The mitogen-activated protein kinase is stimulated by Ang II–AT1–R interaction, further regulating the transcription of some genes: *MMP*, *PAI-1*, *CTGF*, and *TGF-β* [37]. Smad 2/3 phosphorylation, with Smad complex translocation into the cell nucleus—and TGF-β synthesis are also induced by Ang II [37]. Ang II can also increase ROS production and cause cardiac hypertrophy through the Rho G pathway [37]. Sygitowicz et al. reported Ang II-MAPK, TGF-β-Smad signaling pathways, and Rac1-dependent CTGF activation to be the mechanisms in atrial remodeling and fibrosis in AF [37]. Ang II was also demonstrated to have an epigenetic-dependent prohypertrophic effect on atrial cardiomyopathy through the regulation of histone acetylation via the cytoplasmic-nuclear shuttling of HDACs [39]. Thus, the MEF2 binding to the promoter of hypertrophy-related genes is produced. This constitutes a novel mechanism of atrial hypertrophy regulation reported by Zheng et al., which might provide a promising therapeutic strategy for atrial cardiomyopathy [39].

Leukocyte activation is also triggered at the LA level with the subsequent release of inflammatory stimuli [38]. Fibroblast proliferation and differentiation into the myofibroblast phenotype are activated. Thus, the extracellular matrix (ECM) components are released, including fibronectin, procollagen, laminin, elastin, fibrillin, proteoglycans, glycoproteins matrix metalloproteinases (MMPs), and tissue inhibitors of MMPs [38].

Structural remodeling is one of the factors influencing the AF pathophysiology in AS, in addition to ion channel dysfunction, Ca^2+^-handling abnormalities, and autonomic neural dysregulation [40]. Fragão-Marques et al. studied, for the first time, the atrial remodeling in AS patients with chronic AF through fibrosis quantification and target extracellular matrix protein gene expression analysis. In AF patients, an increased collagen type III and decreased TIMP1 and TIMP2 gene expressions were found, accompanied by anincreased cardiomyocyte area and atrial fibrosis discovered during the histologic quantification [40]. The atrial expressions of collagen I, collagen ratio I/III, MMP2, MMP9, MMP16, TGFβ1, and TIMP 4 genes were similar in AF and non-AF patients. The MMP16/TIMP4 ratio was decreased, while serum TIMP1 and TIMP2 were increased in AF patients [40]. In AS, the occurrence of AF may lead to misclassification of severity because it associates lower maximum and mean pressure gradients [40]. Researchers reported that the aortic valve mean gradient was inversely correlated with the MMP2/TIMP1 ratio, collagen type I gene expression, collagen type I/III ratio, and serum TIMP1 levels [40]. Collagen type I gene expression and collagen type I/III ratio were associated with the aortic valve area [40].

## 3. Echocardiographic Assessment of the Left Ventricle and Left Atrium in Aortic Stenosis

### 3.1. Left Ventricular Remodeling and Function

The assessment of LV remodeling and function is mandatory in all AS patients. Measuring the LVEF via the biplane method of discs is always recommended and, whenever feasible, via 3D echocardiography [41]. However, although LVEF carries important prognostic information and is the basis for therapeutic decisions in AS patients, it is not an index of contractility and decreases only late in the course of the disease, being unable to detect early LV dysfunction and trigger initiation of proper treatment. All of this led to a focus not only on the subclinical changes in LV function but also on identifying the correlates of heart failure symptoms in patients with severe AS and preserved LVEF who would benefit from early AVR.

It is known that LV global longitudinal strain (GLS), assessed with speckle tracking echocardiography (STE), is an accurate and widely available marker of early, subclinical LV dysfunction, which occurs before LVEF impairment. The use of advanced echocardiography imaging allows the assessment of myocardial mechanics in terms of displacement, velocity, strain, and strain rate in longitudinal, radial, and circumferential directions. In AS, LV longitudinal strain is impaired, especially in the basal segments (Figure 1). The development of LV fibrosis is responsible for the alteration in GLS, and both are related to valve disease and prior myocardial infarction [42].

The parameters of LV systolic function, assessed via speckle tracking echocardiography (STE), were correlated with the presence of heart failure symptoms in AS. Moreover, GLS has proved useful for risk stratification in asymptomatic severe AS patients with preserved LVEF [43]. Studies have shown that GLS predicts post-operative LV dysfunction and outcomes better than LVEF [44]. GLS improved early after AVR, reflecting its relationship with LV afterload mismatch, as well as fibrosis [44,45].

### 3.2. Left Atrial Remodeling and Function

Echocardiography also enables the assessment of atrial structural and functional changes. Progressive LA enlargement and impairment of all three components of LA longitudinal phasic function have already been demonstrated in patients with severe AS [46]. LA size offers more stable information than Doppler-derived parameters of LV diastolic function (e.g., E/e’, TR velocity), which provide a snapshot of LV diastolic function at the very moment of examination.

Two-dimensional STE is recognized as a powerful, accurate imaging method for assessing the LA phasic function (Figure 2).

Mateescu et al. proposed the LA systolic strain rate, in addition to AVA, as a tool for risk stratification in patients with asymptomatic AS [47]. STE is also clinically useful in severe AS, revealing subtle LA dysfunction in these patients before AVR (Figure 3).

Moreover, impaired LA longitudinal function predicted post-operative AF in patients who underwent AVR for isolated severe AS [48].

However, O’Connor et al. reported that LA dilation does not reflect an intrinsic LA dysfunction, and alteration in LA strain parameters does not perfectly track the increase in LA size in severe AS patients [49]. This finding may be explained by the Frank–Starling mechanism: although initially acting as a compensatory mechanism in response to LA dilation, it can also lead to a deterioration in LA function with time [49]. Additionally, LA dysfunction might appear before LA enlargement and LV damage [38]. Until recently, the main underlying mechanism of LA dysfunction in AS was considered to be the increase in LA afterload through higher LV diastolic pressures [50].

Previous data suggest that the presence of LA MF is independently associated with impairment of LA longitudinal function, with subsequent development of heart failure symptoms and/or AF [51]. Therefore, early detection of LA dysfunction is clinically important for severe AS therapeutic management because once MF has developed, the remodeling process may be irreversible.

Moreover, experimental studies suggest that older age and chronic myocardial stretch may contribute to the development of both LA wall fibrosis and remodeling in severe AS, leading to intrinsic LA myocardial alterations and subsequent functional impairment [52].

## 4. Cardiac Magnetic Resonance in Aortic Stenosis

CMR has the advantage of combining anatomy and function multi-parameters with detailed soft tissue characterization. This is particularly important in AS, as understanding the myocardial damage and function may be clinically relevant, beyond the valve hemodynamics and obstruction. The use of CMR to detect MF is currently the most studied application of myocardial tissue characterization in AS patients [53]. MF is the pathological key driving LV decompensation in AS and the transition from LVH to heart failure. It can be divided into diffuse fibrosis, which occurs earlier and is reversible, and replacement fibrosis, which occurs later and is irreversible.

Late gadolinium enhancement (LGE) CMR is the gold standard imaging method for assessing focal, replacement fibrosis (Figure 4).

The development of LGE in AS appears to be a marker of LV decompensation and predicts further rapid progression of fibrosis [54]. Importantly, studies have shown that this MF does not regress after AVR, and the amount of fibrosis that develops while awaiting AVR persists for life in AS patients [54]. Thus, the more MF, the worse the long-term prognosis [55]. LGE detection in AS patients may offer prognostic information, and performing AVR in patients with severe asymptomatic AS, but who have evidence of early scarring on cardiac MRI, will reduce mortality and lead to improved long-term symptoms following AVR. Optimizing the timing of AVR by LGE in clinical daily practice is being tested in the randomized EVOLVED trial [56].

T1 mapping techniques can provide overall assessments of the extracellular compartment, unlike LGE, which is insensitive to the detection of diffuse interstitial fibrosis. T1 mapping and extracellular volume fraction (ECV%) have prognostic significance across a range of cardiomyopathies. ECV% differentiates between intracellular and extracellular components of the myocardium. The indexed ECV has also been studied in AS, and this measure provides an estimate of the total burden of MF. T1 mapping does not require gadolinium contrast and provides an insight into the early assessment of diffuse fibrosis, which has been proved to reverse following AVR [57].

The assessment of MF by CMR may in the future refine the selection of asymptomatic patients with severe AS who may benefit from early AVR, although this requires confirmation in larger prospective studies. A prevalence of up to 16% of wild-type transthyretin cardiac amyloidosis was reported in patients with severe AS [58]. Apical sparing of longitudinal myocardial deformation has been described in amyloidosis, differentiating cardiac amyloidosis from other causes of LVH. CMR may detect interstitial expansion associated with cardiac amyloidosis, while Tc-99m labeled bone scintigraphy is the imaging test used for the detection of cardiac transthyretin amyloidosis, differentiating it from light-chain cardiac amyloidosis [58]. According to the most recent data, AVR is not futile in these patients, as novel amyloid therapies are now available, and they may still benefit from aortic valve intervention.

## 5. Molecular Biomarkers

Molecular biomarkers related to cardiac fibrosis and remodeling have been tested to find an accurate risk stratification strategy in severe AS patients optimizing the therapeutic decision. Table 1 summarizes the clinically significant studies using biomarkers, alone or in a combined multi-parametric approach in this setting.

### 5.1. Galectin-3

Galectin-3 (Gal-3), a profibrotic and pro-inflammatory molecule, regulates apoptosis through interaction with activated K-Ras protein, acts through the Wnt signaling pathway, and modulates cell–ECM adhesion by binding to the main proteins involved in cell adhesion [59]. Gal-3 specifically binds to cardiac fibroblasts and induces fibroblast proliferation, upregulating cyclin D1 [60] and inducing collagen I production [59,60,61]. Gal-3 expression and collagen production are increased by Ang II through the PKC-α pathway [62].

Ibarrola et al. found that, in AS, the circulating levels of Gal-3 could reflect oxidative stress [62]. Gal-3 downregulated Prx-4 in cardiac fibroblasts and prohibitin-2 expression without modifying other mitochondrial proteins [62]. The same biomarker increased peroxide, nitrotyrosine, malondialdehyde, and N-carboxymethyl-lysine levels, decreasing total antioxidant capacity [62]. In an experimental study, Frunza et al. found that Gal-3 loss may delay the hypertrophic response after pressure overload [63], suggesting direct activation of a hypertrophic program in cardiomyocytes by Gal-3 or Gal-3-mediated modulation of macrophages toward a phenotype, which promotes hypertrophy [63]. In another experimental study, the increase in myocardial Gal-3 expression was associated with cardiac fibrosis and inflammation in short-term AS—changes, that were prevented by Gal-3 blockade [64].

Gal-3 has been studied in clinical cardiac settings, although it is not specific for identifying MF and depends on renal function [65,66,67]. High Gal-3 levels predicted mortality in chronic heart failure (CHF) [68,69] and were associated with a higher risk of MF and the risk of sudden cardiac death [70].

Arangalage et al. do not sustain support the use of Gal-3 in the decision-making process for asymptomatic patients with AS [65]. In their prospective study, which enrolled patients with all grades of AS severity, no association between Gal-3 and the functional status or AS severity was established. Moreover, age, female gender, hypertension, diabetes, reduced LVEF, diastolic dysfunction, and creatinine clearance were the independent determinants of Gal-3 level [65]. GLS and NT-proBNP emerged in another study as the most reliable predictors of major adverse cardiac events (MACEs) in severe AS patients, while Gal-3 performed more poorly [67]. In symptomatic degenerative AS patients with an aortic valve area (AVA) index of 0.4 ± 0.1 cm^2^/m^2^, baseline Gal-3 was unrelated to age, symptomatic status, AVA index, LVEF, LV mass index, or valvulo-arterial impedance, and it was negatively correlated with the estimated glomerular filtration rate (eGFR) (*r* = −0.61, *p* < 0.001) [66]. Gal-3 tended to predict mortality at a cut-off of 17.8 ng/mL, but the result was not maintained after adjustment for eGFR (HR: 1.70 (0.61–4.73), *p* = 0.3) [66].

On the other hand, Baran et al., prospectively studying moderate-to-severe degenerative AS patients over a 48-month period, reported that the CHF exacerbations were mainly dependent on Gal-3 level, correlating with vascular stiffness parameters [71]. White et al.’s systematic review and meta-analysis showed that high baseline Gal-3 levels were significantly associated with all-cause mortality (HR 1.82; 95% CI 1.27–2.61; *p* < 0.001) in AS patients [72]. Gal-3 may also be a valuable prognostic predictor in AS patients with myocardial remodeling, especially when a concentric hypertrophy geometry was developed [73].

Ramos et al. recently proposed an integrative medicine approach with multi-disciplinary information in asymptomatic severe AS patients’ evaluation [74]. The authors reported that, although only NT-proBNP presented significance in the multi-variate analysis predicting MACEs, the combination of NT-proBNP and Gal-3 had powerful stratification capabilities [74]. The levels used in this setting were 14.3 pg/mL for Gal-3 and 435 pg/mL for NT-proBNP [74].

Thus, clinical studies reported contradictory results regarding the role of Gal-3 in AS risk stratification and outcome. When this biomarker was studied in combination with other serum biomolecules, the results were more promising [74]. Giritharan et al.’ study findings are awaited. A group of serum biomarkers containing Gal-3 (BNP, Gal-3, GDF-15, sST2, OPG, miRNA 19b and 21) combined with imaging biomarkers (GLS) are going to be tested as a tool for accurately assessing the risk in AS patients with type 2 diabetes mellitus [75].

### 5.2. sST2

sST2 represents the soluble isoform of the protein suppression of tumorigenicity-2, that inhibits the effects of IL-33/ST2L signaling by acting as a decoy receptor for IL-33 [76]. In parallel, sST2 increases nitrotyrosine, protein oxidation, and peroxide production, enhances the secretion of pro-inflammatory cytokines interleukin (IL)-6, IL-1β, and monocyte chemoattractant protein-1 (CCL-2), and downregulates mitofusin-1 (MFN-1), a protein involved in mitochondrial fusion, in human cardiac fibroblasts [77].

sST2 has been studied in severe AS as a promising biomarker. It represents an independent predictor of cardiovascular events in this setting [78]. Circulating levels of sST2 ≥ 28.8 ng/mL were associated with replacement MF and advanced LVH [79]. Elevated sST2 levels were more often correlated with mid-wall than with subendocardial fibrosis [79]. Levels of sST2 > 23 ng/mL could identify asymptomatic patients who would develop symptoms during follow-up [78]. sST2 was independently related to LA index (*p* < 0.0001) and AOA (*p* = 0.004) in severe AS [78]. This biomarker could be also used to non-invasively assess the pulmonary hypertension (PH) in severe AS [80].

Fabiani et al., on the contrary, reported that, although the severe AV patients presented elevated sST2 levels, no correlation was found between these levels and MF—, assessed by interventricular septum biopsy—[81]. sST2 ≥ 284 ng/mL best discriminated the controls from patients with impaired GLS (GLS ≤ 17%) and increased E/e′ average [81]. Sobczak et al. found that NT-proBNP and sST2 concentrations cannot be used to differentiate the patients according to the severity of AS [82]. However, Mirna et al. recently drew attention to evidence suggesting that different enzyme-linked immunosorbent assay (ELISA) kits could result in diverging serum concentrations measured [83].

Other authors tried to create a model of multiple biomarkers (GDF15, sST2, and NTproBNP) to better perform the risk stratification of patients with AS [84]. These biomarkers were studied separately, together, and when added to the STS score. When added to the STS score, the number of biomarkers elevated provided a category-free net reclassification improvement of 64% at one year (*p* < 0.001) [84].

sST2 is now part of the serum biomolecules proposed by Giritharan et al. to be studied in combination with an imaging biomarker, GLS, in the risk stratification of AS patients [75].

### 5.3. Micro-RNAs

The cardiac fibrosis process is influenced by micro-RNAs (miRNAs) targeting the key molecules, which mediate the transcription of ECM genes and TGFβ signaling [85]. Experimental studies showed that in the pressure-overloaded LV, the TGFβ-dependent regulatory mechanism is involved in miRNA-21 overexpression in cultured fibroblasts [86]. Upregulation of DICER miRNA occurs with its transcript levels correlating directly with TGF-β1, SMAD2, and SMAD3. The interaction between SMAD2/3 and DICER1 further leads to pre-miRNA-21 processing to mature miRNA-21 [86]. miRNA-21 regulates the ERK-MAP kinase signaling pathway involved in the cardiac fibroblast responsible for cardiac hypertrophy, cardiac remodeling, and fibrosis [87]. The profibrotic signaling in the pressure-overloaded heart is also enabled by the downregulation of the miRNA-221/222 family, which may target several genes involved in TGF-β signaling, including *JNK1* (*c-Jun N-terminal kinase 1*), *TGF-β receptor 1* and *TGF-β receptor 2*, and *ETS-1* (*ETS proto-oncogene 1*) [88].

The levels of miRNA-29b [89], miRNA-29c [90], miRNA-210 [91], and miRNA-133a [92] are upregulated in AS, while the plasmatic level of miRNA-22 is not [91]. The increase in miRNA-210 levels in AS patients was comparable to the increase in NT-proBNP levels and was significantly associated with mortality in moderate–severe AS patients [91]. The changes in miRNA-29b targets (collagens and GSK-3β) revealed a remodeling pattern, which was more fibrotic in males and hypertrophic in females [89]. García et al. proposed miRNA-29b as a potential prognostic biomarker, since its pre-operative circulating expression was a significant negative predictor of reverse remodeling after AVR and paralleled the severity of LVH in AS women [89]. Another study proposed miRNA-133a as a prognostic tool because its pre-operative plasma levels predicted the regression potential of LVH after AVR [92]. Adewuyi et al.’s (2022) systematic review observed the paucity of studies concerning miRNAs in AS, the available ones being overall of low-to-moderate quality [87]. The most frequently reported miRNA associated with fibrosis in AS was miRNA-21 [87].

Its myocardial [93] and plasmatic levels [87,93,94,95] were significantly higher in the AS and correlated directly with the echocardiographic mean transvalvular gradients [93,94]. miRNA-21 overexpression was confined to interstitial cells and absent in cardiomyocytes [93], reflecting the presence of significant MF (defined as MF % higher or lower than 10%) [94]. Studies proposed a combined evaluation of miRNA-21, a marker of MF, and LVH, together with advanced echocardiographic imaging [94,95], especially with GLS evaluation [94], in order to better stratify the risk in patients for whom the guidelines do not provide a clear therapeutic strategy. In this setting, Giritharan et al.’s study results are awaitedare awaiting publication. miRNA-19b and 21 will be studied with a multi-parametric approach targeting better risk stratification in AS patients [75].

### 5.4. Growth Differentiation Factor-15 (GDF-15)

Serum growth differentiation factor-15 (GDF-15) is increased by aging and in response to cellular stress and mitochondrial dysfunction [96]. Studies reported a strong relationship between GDF-15 levels and AS severity degree [97,98,99,100]. Elevated GDF-15 levels in severe AS patients are associated with impaired functional capacity, poorer performance on fragility testing, and LV dysfunction [99,100].

GDF15 predicted a low (<5) Katz score, independent of the relationship with LV mass, age, renal function, or indices of LV dysfunction [100]. GDF-15 > 3393 pg/mL, NT-proBNP > 4060 ng/L, LA volume > 100 mL, mitral valve deceleration time (MV DT) ≤ 177.5 ms, E/E’ ratio > 14 predicted PH in patients with severe AS, helping stratify patients for earlier surgical treatment before the development of pulmonary hypertension [101]. GDF-15 and NT-proBNP pericardial levels correlated with atrial matrix remodeling in AF from severe AS [102].

Lindman et al. used a model with three biomarkers (GDF15, sST2, and NTproBNP), which improved the reclassification of severe AS patient risk when added to the STS score [84]. Giritharan et al. proposed to study serum biomarkers (BNP, Gal-3, GDF-15, sST2, osteoprotegerin, miRNA 19b and 21) with an echocardiographic parameter, GLS, in stratifying the risk in severe AS patients with type 2 diabetes mellitus [75]. The authors have not reported the results yet. A future direction starting from Gumauskienė et al.’s results might be the test of the GDF-15/NT-proBNP combination together with LVDD parameters (E/E’ ratio, MV DT, and LA volume) to identify the high-risk asymptomatic severe AS patients [101].

### 5.5. Collagen Turnover Biomarkers, Matrix Metalloproteinases (MMPs), and Tissue Inhibitors of MMPs

Collagens, a superfamily of 28 members, are synthesized as procollagens, cleaved to the mature form, and then enzymatically (MMPs) cleaved and released as biologically active fragments [59]. The balance between MMPs and TIMPs maintains the homeostasis of ECM. Propeptides reflect the synthetic process, whereas the degradation epitopes reflect the fibrotic process [59]. In the cardiac setting, the amino-terminal peptide of procollagen type I (PINP), C-terminal peptide of procollagen type I (PICP), and amino-terminal peptide of procollagen type III (PIIINP) have been studied as neoepitopes (circulating collagen fragments), while the carboxy terminal telopeptide of collagen I (CITP) has been studied as a degradation-related epitope.

High serum PICP and PIIINP levels were significantly associated with MF in some studies [103], although others found that circulating collagen metabolites are not reliable surrogate MF measures in AS [104]. Nevertheless, CITP and PIIINP were strongly associated with heart failure and LV dysfunction in AS patients [104]. LV systolic longitudinal strain and diastolic blood pressure were independent predictors of plasma PIIINP in AS patients with normal LVEF [105]. Foussier et al. reported no significant associations between global ECV and the expression of PICP, CITP, Gal-3, and PIINP [106]. Global ECV was poorly correlated with TIMP-1 and MMP-2, while CITP was moderately correlated with EDVi, ESVi, and myocardial mass [106]. PICP was poorly correlated with EDVi and ESVi [106]. Lange et al., studying the functional and structural reverse myocardial remodeling following TAVI, reported a significant inverse correlation between the CITP:MMP1 ratio and non-ischemic LGE volume, and no statistical association of the CITP:MMP1 ratio with LV matrix volume [107].

The dysregulation of myocardial MMPs and TIMPs starts at an early disease stage, when the LV function is still normal in AS [108,109]. In mild AS, the LV end-diastolic volume index was significantly associated with MMP-1; the aortic valve mean pressure gradient was independently associated with MMP-2; MMP-2 was significantly associated with TGF-β and IL; and IL-1 was independently associated with TIMP-1 [109]. Studies demonstrated, based on MMP-1 levels, that if the inflammatory process was related to the mild stage of AS, the most prominent extracellular matrix remodeling occurred in moderate AS [110]. MMP-1 presented the highest level of MMP-1 in moderate AS patients [110]. Its level correlated positively with MMP-9, not with MMP-3 [110]. TIMP-1 was upregulated in compensated hypertrophy and presented the highest level in failing hearts [111]. The cardiac expression of TIMP-1 and TIMP-2 was significantly increased in chronic pressure-overloaded human hearts and was related to the degree of interstitial fibrosis [103].

Bäz et al. reported that B+/C+ Tenascin-C and MMP-9 changes correlated with LA involvement in severe AS, while TIMP-1, ED-A+ Fn, ET-1, and NGAL changes were associated with the transition to PH with right heart dysfunction [112]. MMP-3 correlated with LV end-diastolic dimension [113]. AS patients with AF presented increased collagen type III synthesis, decreased MMP16/TIMP4 ratio, and increased serum TIMP1 and TIMP2 proteins— [40]. An increase in MMP-2 and a decrease in TIMP-2 and -4 are found in compensatory heart disease, and an increase in MMP-9 and TIMP-3 in decompensated states [114]. The serum levels of MMP-9, MMP-3, and TIMP-1 were proposed to evaluate PH in severe AS [115]. MMP-28 level was statistically significantly correlated with the peak blood flow velocity and mean pressure gradient of the transaortic valve in severe AS [116].

### 5.6. Brain Natriuretic Peptide, N-Terminal-Pro-Brain Natriuretic Peptide, and Troponin

Brain natriuretic peptide (BNP) is mainly synthesized by LV myocytes as a response to pressure or volume overload. Its precursor is cleaved in BNP, the active amino acid, and N-terminal proBNP (NT-proBNP), an inert amino acid. Clinicians must be aware that the half-life of the most used natriuretic peptides is different: BNP’s half-life is 20 min, while NT-proBNP’s half-life is 90–120 min [117]. BNP synthesis is mainly dependent on p38 MAPK, a subtype of mitogen-activated protein kinase MAPK, which is activated through the integrin by mechanosensors. p38α induces BNP gene transcription through activator protein-1 (AP-1), while p38β regulates BNP gene expression through endothelin-1 (ET-1)-induced transcription factor GATA-4 [118]. Additionally, angiotensin II (Ang II) and ET-1 complexes are stimulated by the pressure/volume overload further activating the BNP gene via p38 MAPK and extracellular signal-regulated kinase (ERK) signaling pathways [118].

The natriuretic peptides help perform risk stratification for AS patients regarding heart failure, syncope, and sudden cardiac death [119,120,121]. NT-proBNP independently predicted symptom-free survival in asymptomatic severe AS patients [121,122]. Studies reported that asymptomatic severe AS patients with BNP levels of <100 pg/mL had relatively low event rates and might be safely followed with a watchful waiting strategy [123].

A higher NT-proBNP ratio NT-proBNP divided by the upper limit of normal NT-proBNP for age and sex) predicted mortality in more valvular heart diseases with the strongest association detected for AS [120]. A higher baseline NT-proBNP ratio and a significantly reduced LV-GLS were related to LV asymmetric remodeling, found in ~20% of mild or moderate AS patients [124]. The prognostic value of the NT-proBNP ratio was observed in patients with severe or non-severe AS/aortic regurgitation and those treated with early AVR as well [125].

Weber et al. calculated a baseline NT-proBNP cut-off of 640 pg/mL, which was discriminative for an adverse outcome in severe AS patients, especially in the conservative treatment approach [126]. Higher NT-proBNP levels (median NT-proBNP 888 pg/dL) significantly predicted mortality in moderate AS patients— [127]. Gumauskienė et al. calculated an NT-proBNP cut-off value of NT-proBNP of 4060 ng/L for predicting PH in severe AS patients, considering that the normal range of NP is inappropriate for this group of patients with chronically elevated LV filling pressures [101]. Studies reported that pericardial NT-proBNP had a higher diagnostic accuracy for AF in AS than its serum level. Together with pericardial GDF-15, it could predict AF in AS and correlate with atrial matrix remodeling [102].

High-sensitivitytroponin T (hsTnT) adds information to NT-proBNP as a routinely available biomarker for risk stratification concerning post-operative survival in patients with severe AS admitted for AVR [128]. Cardiac troponins are released mainly due to myocyte necrosis, apoptosis, and myocyte cell turnover. Auensen et al. reported that NT-proBNP, hsTnT, and hs-CRP did not have prognostic value concerning all-cause mortality following AVR and that hsTnT was independently associated with MACE after AVR [129]. Although high serum cTnT and NT-proBNP are common in AS as measures of maladaptive remodeling and cardiac injury, these biomarkers can predict post-TAVI mortality better than the LV mass index [16].

hsTnT level > 14 pg/mL was associated with ischemic coronary events risk in mild-to-moderate– AS patients [130], while hs-TnT > 10 ng/L was associated with a high risk of events within 12 months in asymptomatic severe AS patients [131]. Elevated hs-TnT levels were associated with shorter time to surgery and with increased myocardial mass in AS patients, indicating that hs-TnT could be a potential biomarker for determining the time of surgical intervention in AS patients [132]. Clinicians must be aware that both natriuretic peptides and troponins may be elevated due to other extracardiac diseases, such as cancer [133].

Starting from their multi-centric study results, which found the cTnT, NT-proBNP, and GLS combination to be a prognostic tool in symptomatic severe AS patients undergoing TAVI, Perry et al. proposed the same approach for asymptomatic severe AS patients [134]. Some researchers proposed the use of a multi-parametric approach—LVDD parameters (E/E’ ratio, MV DT, and LA volume) and biomolecules (GDF-15 and NT-proBNP)—in order to find the high-risk asymptomatic severe AS patients who would benefit from earlier surgery [101]. Others advanced the idea of studying another group of serum biomarkers (GDF15, sST2, and NT-proBNP) and the STS score to assess the risk in AS patients [84]. Giritharan et al. study might bring new data in this field [75].

### 5.7. Other Biomarkers

Annexin A1, an endogenous anti-inflammatory mediator, was studied in the fibrosis context. In AS, its pericardial level predicted AF in severe AS patients [135]. Activin A, a member of the transforming growth factor beta superfamily, correlated negatively with the physical performance after AVR [113]. C-type natriuretic peptide (CNP), an endothelial product, was superior to NT-proBNP for TAVI risk evaluation, especially in patients with LVEF < 50% [136]. The mid-regional proadrenomedullin (MR-proADM) was also found to predict all-cause mortality, heart failure hospitalization, and progression to NYHA class III–IV in moderate–severe AS patients, especially when combined with hsTnT or with NT-proBNP [137].

Osteopontin (OPN) can induce the activation of p90 ribosomal s6 kinase, Akt, glycogen synthase kinase-3β, NFAT/GATA-4, calcineurin-NFAT, and serum- and glucocorticoid-inducible kinase as a response to pressure or volume overload [138]. Baseline OPN levels were associated with adverse outcomes in severe AS patients undergoing TAVI [139]. The pre-operative OPN levels can predict the type of LV hypertrophy. Low pre-operative levels were predictors of advanced LV myocardial regression after AVR, while high levels were associated with eccentric and less reversible LVH [140]. LVH regression was also related to transforming growth factor-beta1 levels [141].

Angiotensin-converting enzyme 2 (ACE2) activity was also studied in the AS context. Although the presence of hypertension and/or antihypertensive therapies can introduce some bias in the analysis, researchers reported a positive correlation between plasma ACE2 activity and LV mass in AS patients. ACE2 was proposed as a marker of early myocardial decompensation in AS because its high plasmatic levels were associated with increased LV diastolic volume but not with LVEF or GLS [142]. Copeptin (C-terminal pro-vasopressin), a surrogate marker of the arginine–vasopressin system, was proposed by Yalta as part of the decision algorithm for asymptomatic severe AS patients [143].

**Table 1 biomolecules-13-01661-t001:** Studies of the main biomolecules as potential biomarkers for risk stratification in aortic stenosis.

Biomarker	Authors	Year	Reference	Biomarker	Authors	Year	Reference
Gal-3	Arangalage et al.	2016	[65]	MMPs	Liu et al.	2004	[115]
	Zhou et al.	2016	[73]		Fielitz et al.	2004	[108]
	Bobrowska et al.	2017	[66]		Bjørnstad et al.	2008	[113]
	Agoston-Coldea et al.	2018	[67]		Givvimani et al.	2010	[114]
	White et al.	2021	[72]		Zhou et al.	2020	[116]
	Baran et al.	2022	[71]		Bäz et al.	2020	[112]
	Ramos et al.	2023	[74]		Park et al.	2014	[109]
					Lurins et al.	2019	[110]
sST2	Lancellotti et al.	2015	[78]		Foussier et al.	2021	[106]
	Lindman et al.	2015	[84]				
	Fabiani et al.	2017	[81]	TIMP	Fielitz et al.	2004	[108]
	Sobczak et al.	2017	[82]		Liu et al.	2004	[115]
	Boxhammer et al.	2022	[80]		Givvimani et al.	2010	[114]
	Arrieta et al.	2023	[79]		Park et al.	2014	[109]
					Bäz et al.	2020	[112]
NT-proBNP	Bergler-Klein et al.	2004	[122]		Foussier et al.	2021	[106]
	Weber et al.	2006	[126]				
	Lindman et al.	2015	[84]				
	Sobczak et al.	2017	[82]	Collagen turnover biomarkers	Du et al.	2012	[105]
	Agoston-Coldea et al.	2018	[67]		Kupari et al.	2013	[104]
	Gumauskienė et al.	2018	[101]		Foussier et al.	2021	[106]
	Ito et al.	2020	[127]		Zhang et al.	2022	[103]
	Zhang et al.	2020	[120]				
	White et al.	2021	[72]	miRNAs	Villar et al.	2013	[96]
	Tan et al.	2022	[137]		García et al.	2013	[92]
	Perry et al.	2022	[134]		Røsjø et al.	2014	[91]
	Bernard et al.	2023	[125]		Derda et al.	2015	[90]
	Ramos et al.	2023	[74]		Fabiani et al.	2016	[94]
					García et al.	2020	[89]
BNP	Bergler-Klein et al.	2004	[122]				
	Lancellotti et al.	2015	[78]				
	Nakatsuma et al.	2019	[123]				

hs-TnT	Ferrer-Sistach et al.	2019	[131]	GDF-15	Lindman et al.	2015	[84]
	Barbieri et al.	2019	[128]		Gumauskienė et al.	2018	[101]
	Holmgren et al.	2020	[132]		Fabiani et al.	2020	[100]
	White et al.	2021	[72]		Hofmanis et al.	2021	[98]
	Baran et al.	2022	[71]		Basmadjian et al.	2023	[99]
	Tan et al.	2022	[137]				
	Perry et al.	2022	[134]				
	Hadziselimovic et al.	2023	[130]				

Relevant clinical studies are presented, focusing on the association between biomarkers and severe aortic stenosis from the risk assessment point of view. Abbreviations: BNP, B-type natriuretic peptide; Gal-3, galectine 3; GDF-15, growth differentiation factor 15; hs-TnT, high-sensitivity cardiac troponin T; miRNAs, MicroRNA; MMPs, metalloproteinases; NT-proBNP, N-terminal prohormone of brain natriuretic peptide; sST2, soluble suppression of tumorigenicity 2 ; TIMP, tissue inhibitors of metalloproteinases.

## 6. Conclusions and Future Perspectives

AS is a disease both of the valve and the myocardium, characterized by fibrosis and calcification of valve leaflets, progressive LV hypertrophy, and myocardial fibrosis. Therefore, a comprehensive assessment of patients with AS is recommended. Beyond the conventional assessment of LV remodeling and ejection fraction, the assessment of LV GLS via STE and myocardial fibrosis estimated via CMR will be increasingly used in the decision-making process in patients with AS in the near future. Such a multi-parametric approach, including biomarkers, can have a role in defining the optimal timing for intervention in apparently asymptomatic severe AS patients and stratifying the risk in patients undergoing AVR. However, confirmatory data from randomized clinical trials are awaitedare awaiting publication in order to define their utility in current clinical practice and incorporate them into the management algorithms. A better knowledge of the underlying mechanisms of the transition from LV hypertrophy to LV decompensation may provide an insight into novel mediators of cardiac remodeling and decompensation for possible diagnostic and prognostic use in AS and also identifying the biotargets for novel pharmacological therapies.

## Figures and Tables

**Figure 1 biomolecules-13-01661-f001:**
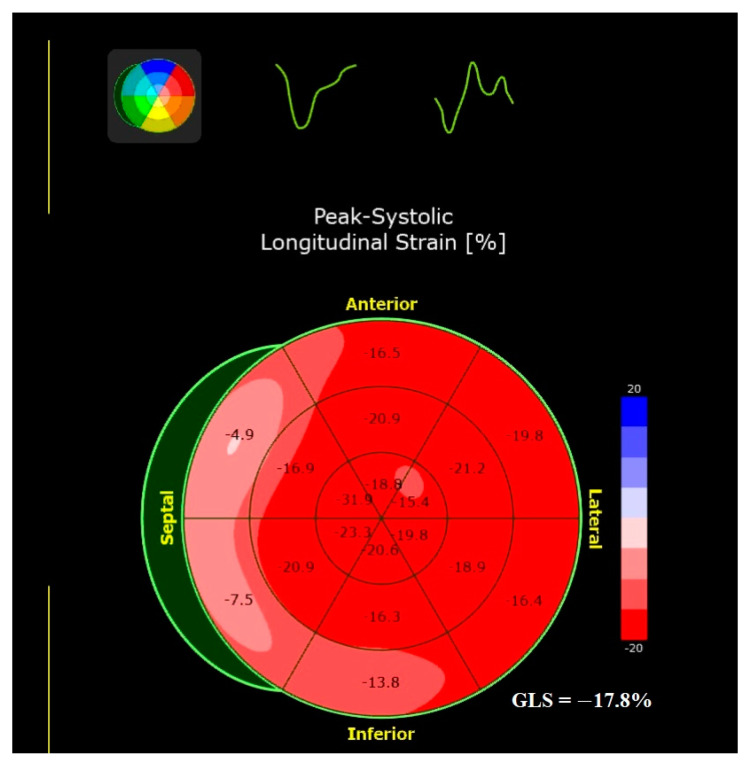
Left ventricular global longitudinal strain measured via speckle tracking echocardiography in an asymptomatic patient with severe aortic stenosis. The LVEF was normal (64%), and the mean transvalvular gradient was 48 mm Hg. The patient had concentric LV hypertrophy, more prominent in the interventricular septum. An impaired LV GLS (−17.8%) was found in this patient, with more severely reduced values of longitudinal deformation in the basal segments of the interventricular septum. Coronary angiography revealed no significant coronary artery disease. Abbreviations: LV, left ventricle; GLS, global longitudinal strain; LVEF, left ventricular ejection fraction.

**Figure 2 biomolecules-13-01661-f002:**
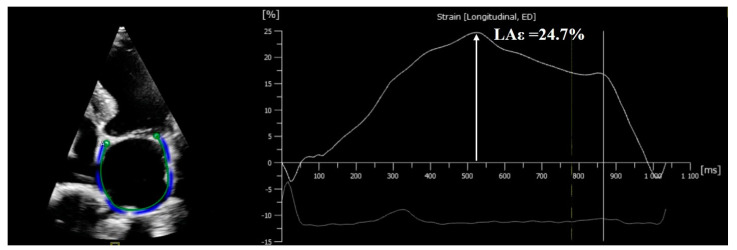
Left atrial longitudinal deformation in a 71-year-old patient with asymptomatic severe AS, preserved LVEF, and sinus rhythm. We observe in the right panel the LA longitudinal strain curve with a normal mean value of 24.7% (LAƐ). Abbreviations: AS, aortic stenosis; LVEF, left ventricular ejection fraction; LA, left atrium.

**Figure 3 biomolecules-13-01661-f003:**
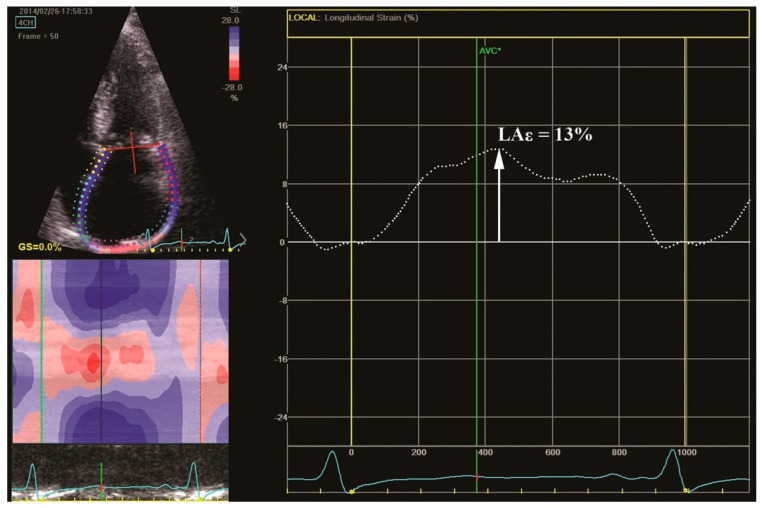
Left atrial longitudinal deformation in a 73-year-old patient with symptomatic severe AS, preserved LVEF, and sinus rhythm. We observe the LA longitudinal strain curve with a reduced mean value of 13% (LAƐ). Lower values of LA longitudinal strain (LA reservoir function) are observed in the symptomatic patient with severe AS compared to the asymptomatic one (Figure 2). Abbreviations: AS, aortic stenosis; AVC*, aortic valve closure; LVEF, left ventricular ejection fraction; LA, left atrium.

**Figure 4 biomolecules-13-01661-f004:**
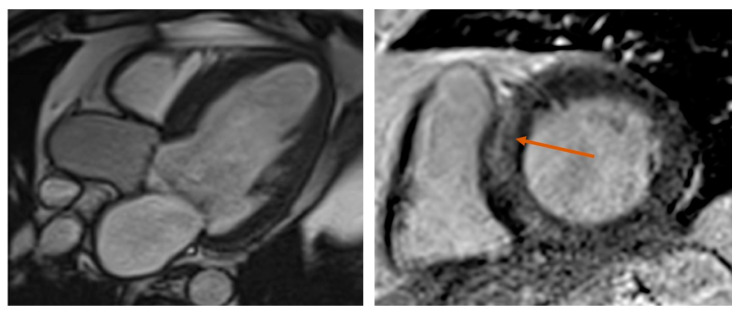
Late gadolinium enhancement on cardiac magnetic resonance imaging. The left and right panels represent LV three-chamber and, respectively, short axis CMR views of a patient with symptomatic severe aortic stenosis and moderate LV global systolic dysfunction (LVEF 38%). The LV was severely dilated, with thin walls, except for the IVS, which was hypertrophied. Coronary angiography revealed no significant coronary artery disease. The orange arrow indicates focal, non-ischemic (mid-wall) LGE in the IVS. Abbreviations: LGE, late gadolinium enhancement; CMR, cardiac magnetic resonance; LV, left ventricle; IVS, interventricular septum; LVEF, left ventricular ejection fraction.

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
