# Peer review of "Novel Biomarkers and Advanced Cardiac Imaging in Aortic Stenosis: Old and New"

_biomolecules, 2023, doi:10.3390/biom13111661_

Round 1

Reviewer 1 Report

Comments and Suggestions for Authors

The manuscript is well structured and with a good analysis of all biomarkers.

However, it is a little lacking in figures.

Therefore it is advisable to include a case of aortic stenosis on MRI if possible.

It is also advisable to include a small table with all the biomarkers treated according to the reference methods.

Comments on the Quality of English Language

The article is well written, minor linguistic revisions are required

Reviewer 2 Report

Comments and Suggestions for Authors

This review focuses on the pathophysiological processes of aortic stenosis (AS), as well as potential imaging indicators and biomarker changes in these processes. The author’s work has great reference significance for future clinical and scientific research, but there are some problems as follows:

1. The manuscript length is too long. The content of each part is too much, resulting in the focus is not prominent. Please reduce some content appropriately. For example, the pathophysiology of part 2 has many overlaps with the imaging section below.

2. Biomarkers parts also have similar problems mentioned above. Taking Gal-3 AS an example, the structure and expression process of Gal-3 are over-elaborated, but the focus should actually be placed on the association between GAL-3 and AS, as well as the future research direction. Besides, a brief summary should be made in the end of each part.

Comments on the Quality of English Language

There is a problem in the ranking of the subtitle of the manuscript, MRI part and biomarker part are both serial number 4.

The sentence in lines 149-150 is repeated twice.

Reviewer 3 Report

Comments and Suggestions for Authors

The authors reviewed pathophysiology, echocardiography and CMR assessment and diagnostic and prospective  biomarkers of aortic stenosis (AS). At first glance it is an interesting and contemporary review. The major consideration is that the review has a very broad scope which from educational point-of-view might be a good idea. However it is always at the expense of a some more in-depth discussion. After reading the manuscript the reviewer has a number of comments/questions. In general there is technical sloppiness at several parts.

1. What is the meaning of a 6 pages long (p. 14-20) Table with only authors' names, years and a one sentence summary? Such strategy lacks a clear message for the readership. Multiple tables organized by biomarker (Gal-3, sST2, GDF15) or group of biomarkers (natriuretic peptides, troponins, collagen turnover, miRNAs) would be a more rewarding strategy for the readership.

2. A number of parameters ( e. g. related to fibrosis) have their gold standard measured by CMR. Therefore an image of a LGR CMR might also be in place.

3. At several places there is a duplication of the text e. g. Lines 291-295, 427-432 and so on.

4. References should be thoroughly revised. In a large number of references there is no correspondence between the numbering in the text and the references list! References are not always in the order of appearance into the text e. g. Ref 113 follows 89 in lines 542-646. What is ref 172 on line 434 or ref 167 on line 473? 
5. A few typing errors e.g. line 475 "galactic".

Comments on the Quality of English Language

None

Round 2

Reviewer 3 Report

Comments and Suggestions for Authors

The authors have answered my questions and thoroughly revised their manuscript. It is beyond any doubt that the quality of the manuscript has dramatically improved.

Comments on the Quality of English Language

Only minor editing required.